# Implementation research on management of sick young infants with possible serious bacterial infection when referral is not possible in Jimma Zone, Ethiopia: Challenges and solutions

Melkamu Berhane[1]*, Tsinuel Girma[1], Workneh Tesfaye[1], Nega Jibat[2], Mulumebet Abera[3], Sufian Abrahim[4], Samira Aboubaker[5], Yasir Bin Nisar[6], Shamim Ahmad Qazi[5], Rajiv Bahl[6], Alemseged Abdissa[7]

1 Department of Pediatrics and Child Health, Jimma University, Jimma, Ethiopia, 2 College of Social Sciences and Humanities, Jimma University, Jimma, Ethiopia, 3 Department of Population and Family Health, Jimma University, Jimma, Ethiopia, 4 Oromia Regional Health Bureau, Addis Ababa, Ethiopia, 5 Department of Maternal, Newborn, Child, Adolescent Health and Ageing, World Health Organization (Retired) Currently WHO consultants, Geneva, Switzerland, 6 Department of Maternal, Newborn, Child and Adolescent Health and Ageing, World Health Organization, Geneva, Switzerland, 7 School of Medical Laboratory Sciences, Jimma University, Jimma, Ethiopia

* melkamuarefayine@gmail.com

**Data Availability Statement:** All relevant data are within the manuscript.

## Abstract

### Introduction

Of 2.5 million newborn deaths each year, serious neonatal infections are a leading cause of neonatal death for which inpatient treatment is recommended. However, many sick new-borns in sub-Saharan Africa and south Asia do not have access to inpatient care. A World Health Organization (WHO) guideline recommends simplified antibiotic treatment at an out-patient level for young infants up to two months of age with possible serious bacterial infection (PSBI), when referral is not feasible. We implemented this guideline in Ethiopia to increase coverage of treatment and to learn about potential facilitating factors and barriers for implementation.

### Methods

We conducted implementation research in two districts (Tiro Afata and Gera) in Jimma Zone, Ethiopia, to learn about the feasibility of implementing the WHO PSBI guideline within a programme setting using the existing health care structure. We conducted orientation meetings and policy dialogue with key stakeholders and trained health extension workers and health centre staff to identify and manage sick young infants with PSBI signs at a primary health care unit. We established a Technical Support Unit (TSU) to facilitate programme learning, built health workers' capacity and provided support for quality control, monitoring and data collection. We sensitized the community to appropriate care-seeking

**Funding:** The study was funded by the Bill and Melinda Gates Foundation through a grant to the World Health Organization. The funder had no role in study design, data collection and analysis, decision to publish, or preparation of the manuscript.

**Competing interests:** Rajiv Bahl and Yasir Bin Nisar are staff members of the World Health Organization. The expressedviews and opinions do not necessarily express the policies of the World Health Organization. This does not alter our adherence to PLOS ONE policies on sharing data and materials.

and supported the health care system in implementation. The research team collected data using structured case recording forms.

## Results

From September 2016 to August 2017, 6185 live births and 601 sick young infants 0–59 days of age with signs of PSBI were identified. Assuming that 25% of births were missed (total births 7731) and 10% of births had an episode of PSBI in the first two months of life, the coverage of appropriate treatment for PSBI was 77.7% (601/773). Of 601 infants with PSBI, fast breathing only (pneumonia) was recorded in 432 (71.9%) infants 7–59 days of age; signs of clinical severe infection (CSI) in 155 (25.8%) and critical illnessin 14 (2.3%). Of the 432 pneumonia cases who received oral amoxicillin treatment without referral, 419 (97.0%) were successfully treated without any deaths. Of 169 sick young infants with either CSI or critical illness, only 110 were referred to a hospital; 83 did not accept referral advice and received outpatient injectable gentamicin plus oral amoxicillin treatment either at a health post or health centre. Additionally, 59 infants who should have been referred, but were not received injectable gentamicin plus oral amoxicillin outpatient treatment. Of infants with CSI, 129 (82.2%) were successfully treated as outpatients, while two died (1.3%). Of 14 infants with critical illness, the caregivers of five accepted referral to a hospital, and nine were treated with simplified antibiotics on an outpatient basis. Two of 14 (14.3%) infants with critical illness died within 14 days of initial presentation.

## Conclusion

In settings where referral to a hospital is not feasible, young infants with PSBI can be treated on an outpatient basis at either a health post or health centre, which can contribute to saving many lives. Scaling-up will require health system strengthening including community mobilization.

## Registration

Trial is registered on Australian New Zealand Clinical Trials registry (ANZCTR) ACTRN12617001373369.

## Introduction

In 2018, an estimated 5.3 million deaths of under-five children occurred, 2.5 million (47%) of them in the first month of life, known as neonatal deaths [1]. Globally, the proportion of neonatal mortality increased from 40% in 1990 to 47% in 2018, whereas it was 36% for sub-Saharan Africa [1]. In Ethiopia, the under-five, infant and neonatal mortality rates were 55, 43 and 30 per 1000 live births, respectively, in 2019 [2]. Neonatal deaths are primarily due to preterm birth complications, intrapartum complications and infections [3–6].

World Health Organization (WHO) recommends that infants with serious infections and sepsis be treated in a hospital with a combination of injectable gentamicin and penicillin/ampicillin for at least seven days [7]. Many infants with PSBI in low- and middle-income countries do not have access to the recommended inpatient treatment, prompting the need for strategies for home or first-level facility-based management in settings with a high burden of neonatal

deaths [8–11]. Hence,WHO launched a guideline on managing PSBI in young infants 0–59 days of age when referral is not feasible in 2015 [12]. Data from large community-based trials from Africa and Asia contributed to the evidence base for the WHO guideline [13–16]. However, delivering these regimens effectively in diverse settings and existing healthcare systems is challenging.

To address this challenge, we conducted research on implementing the WHO guideline [12]. The primary objective was to identify at least 80% of sick young infants with PSBI in the study districts overa one-year period and ensure that at least 80% of the identified infants received appropriate treatment. The secondary objective was to identify challenges and opportunities (enabling factors) for the implementation at scale of the simplified antibiotic regimen for PSBI when referral is not feasible.

## Methods

Ethiopia has a decentralized three-tier health care system of primary, secondary and tertiary level care characterized by a Primary Health Care Unit (PHCU) composed of five satellite health posts, one health centre and one primary hospital. Above the PHCU are either general or specialized hospitals. A PHCU serves a population of up to 100000, while general and specialized hospitals each serve up to 1.5 and 5 million people, respectively [17–19]. See Box 1 for more details.

### Box 1. Context and setting of the study [17–19]

A health centre is staffed with a team of mid-level health professionals including public health officers, nurses, midwives, environmental health experts, pharmacists and laboratory technicians. A health centre provides comprehensive primary health care which includes promotive, preventive and curative services. One health centre supervises and receives referrals from five satellite health posts.

The Health Extension Programme (HEP) is a programme with its deep root in the community through which several preventive and selected curative services are provided to the community under the 16 essential health packages and within the umbrella of PHCU. The HEWs constitute the core of the HEP, whereas other key actors include model households, the health development army (HDA), the community and the government, which also play significant roles in the implementation of the HEP. Model households are those households that are trained in the HEP packages, implementing these packages after the training, and able to influence their neighbours to adopt the same practices.

A health post is an operational centre for two female health extension workers (HEWs) serving one kebele, which is the lowest administrative unit in Ethiopia and is comprisedof approximately 1000 households or 5000 people. Each HEW is required to spend 75% of her time conducting outreach activities in her respective kebele, and 25% of her time at a health post. All HEWs have completed high school, received additional training for one year on 16 health packages, including maternal and child health (MCH), and are employed in the government health system.

The MCH services provided by HEWs include i) identification and counselling of pregnant mothers; ii) linking to or providing antenatal care; iii) encouraging institutional deliveries; iv) carrying out birth surveillance, and v) providing postnatal care for the

mother-infant pair. HEWs also provide integrated community case management (iCCM) targeting common childhood illnesses.

Since 2013, Community-based Newborn Care (CBNC) was introduced to the health extension programme (HEP) package in which, beyond routine birth and pregnancy surveillance, the HEWs are expected to provide newborn care including identifying and referring sick young infants to higher-level health facilities. Since 2016, HEWs have also been trained to assess, classify and treat young infants with PSBI when the referral is not feasible. Similar to the health centre staff, the HEWs lead community-based activities such as community mobilization and public health campaigns.

Since its establishment in 2003, the Ethiopian HEP has achieved several successes in the areas of maternal, neonatal and child health and several other preventive aspects of community health. Remarkable achievements have been obtained in the areas of family planning, immunization, antenatal care (ANC), malaria prevention and control, TB/HIV prevention and control as well as treatment of common childhood illnesses like diarrheal diseases and acute respiratory tract infections (ARI). Additionally, through the HEP, significant improvements have been demonstrated concerning service utilization, community's knowledge and care-seeking, and latrine construction and utilization [17, 19].

Health Development Army (HDA) is an organized movement of communities forged through participatory learning and action meetings which are designed to improve the implementation capacity of the health sector by engaging communities to identify local challenges, find solutions to these challenges and facilitates scaling up best practices. A functional HDA requires the establishment of health development teams that comprise up to 30 households residing in the same neighbourhood which is further divided into smaller groups of six members, the one-to-five networks. Leaders of the health development teams and one-to- five networks are selected by their team members.

In Ethiopia, the management of PSBI when referral is not feasible is provided under the umbrella of the CBNC programme. The HEWs refer sick young infants with any sign of PSBI to the nearest health centre, which in turn refers the infant to a nearby hospital if PSBI is confirmed. If referral isnot accepted, the HEWs treat the sick young infants with oral amoxicillin and injectable gentamicin for seven days, whereas the health centre staff treat such cases with injectable ampicillin and gentamicin for seven days (Box 2).

### Box 2. Classification and treatment of PSBI according to WHO guideline [12]

PSBI is defined as a young infant 0–59 days old presenting with any of the following signs: fast breathing (respiratory rate $\geq$ 60 breaths per minute), severe chest indrawing, no movement at all or movement only when stimulated, not able to feed at all or not feeding well/stopped feeding well, convulsions, high body temperature ($\geq$38˚C) or low body temperature ($<$35.5˚C).

Classification of PSBI:

Fast breathing pneumonia–infant 7–59 days old presenting with only fast breathing (60 or more breaths per minute)

- Clinical Severe Infection (CSI)–infant 0–59 days old presenting with one or more of the following signs: not feeding well/stopped feeding well, severe chest indrawing, high (38˚C or above) or low (less than 35.5˚C) body temperature, movement only when stimulated, or fast breathing (60 or more breaths per minute) in infants 0–6 days of age (*this last component was added to CSI as a local adaptation by Government of Ethiopia, instead of having it separate as severe pneumonia as designated in WHO guideline*), and

- Critical Illness–infant 0–59 days old presenting with one or more of the following signs: not able to feed at all, convulsions or no movement at all.

Treatment

- Fast breathing only pneumonia in infants 7–59 days old—treat with oral amoxicillin for 7 days without a referral, mandatory follow-up at health posts or health centres on day 4.

- CSI—refer to a hospital. If referral is not feasible to treat with injectable gentamicin daily for 2 days and oral amoxicillin for 7 days, or injectable gentamicin daily for 7 days and oral amoxicillin for 7 days. Counsel family at each occasion on home care and follow-up. Mandatory follow-up at the health post or health centre on day 4.

- Critical illness–refer urgently to a hospital. If a referral is not possible, treat with daily injectable gentamicin and twice-daily ampicillin until referral is possible or for up to 7 days. Counsel family at each occasion on referral and home care.

Ethiopia recently adopted the policy of treatment of sick young infants with CSI signs when a referral is not feasible with twice-daily oral amoxicillin and once-daily injectable gentamicin for seven days (14 doses of amoxicillin and seven injections of gentamicin). This treatment includes infants 0–6 days old presenting with fast breathing only, which is a little different than the WHO guideline where it is a separate category [12]. The Ethiopian Ministry of Health was interested in evaluating the two-day gentamicin regimen, which was an option recommended by the WHO for infants presenting with CSI (Box 2). Hence, Tiro Afata District was selected to implement the two-day injectable gentamicin plus seven-day oral amoxicillin regimen while Gera District was selected to implement the seven-day injectable gentamicin plus seven-day oral amoxicillin regimen. Fast breathing only in infants 7–59 days old was treated with twice-daily oral amoxicillin for seven days in both districts.

## Study sites

Tiro Afata and Gera were selected from the 20 districts (*woreda*) of Jimma Zone, in consultation with the Jimma Zone Health Department, Oromia Regional Health Bureau (RHB) and the implementing partner, John Snow Inc. Last 10 Kilometres (JSI/L10k) Project. Tiro Afata has a population of 152238 with 23 health posts, five health centres and 50 HEWs. Gera has a population of 143555 served by 29 health posts, five health centres and 55 HEWs. There are no hospitals in the selected districts, but two primary hospitals and one specialized hospital in the surrounding districts are referral facilities.

## Study design

In this implementation research, we prospectively collected quantitative and observational data at different levels.

## Study population

The population comprises sick young infants up to 2 months of age with any sign of PSBI. We collected data from all the health posts and health centres in the two districts. However, we were unable to collect information from the referral hospitals in the surrounding districts.

## Interventions

The interventions included policy dialogue, standardization of treatment protocols at the health centre and health post levels, training of HEWs and health centre staff, provision of necessary supplies and commodities at the beginning of implementation in collaboration with JSI/L10K, provision of monthly supportive supervision and quarterly review meetings with the responsible stakeholders. Community sensitization and awareness campaigns were also carried out. A Technical Support Unit (TSU) was established to provide technical back-up to the district health offices and health workers. In collaboration with the implementing partner, the TSU facilitated learning by doing and the replenishing of necessary commodities for health facilities when needed.

For the management of sick young infants, which was considered part of routine HEW activity, HEWs were asked to complete various case recording forms developed by the study team. There was no additional payment given to the HEWs for their routine work. However, a small payment (around US$ 20 per month) was made to the district health office and health centre staff for extra activities that included additional supportive supervision and data collection.

## Data management and analysis

For the quantitative study, data were entered into Epidata version 3.1 and then exported to and analyzed using STATA version 12.0. Descriptive statistics (proportion/percentage) were calculated which included the proportion of sick young infants identified at different levels (health posts and health centres), the proportion of infants with different classifications (fast breathing pneumonia, CSI and critical illness), the proportion of infants referred to higher-level health facilities, the proportion of infants whose caregivers accepted the referral, proportion of infants completing treatment, etc.

## Quality control and assurance

To ensure the quality of the services provided and data collected, the TSU trained HEWs, health centre and district health office staff, as well as the study coordinators, at the beginning and the mid-point of the study. Additionally, the TSU conducted regular monthly supervision and quarterly review meetings with the HEWs and health centre staff and used the meetings to share progress and best practices, challenges and options for overcoming barriers to implementation.

## Phases of the research

The implementation research was carried out in phases.

**1. Orientation and policy dialogue phase.** At the national level, orientation and policy dialogueworkshops were held with the assistance of WHO. They involved all stakeholders

working on newborn health in Ethiopia including the WHO, UNICEF, Save the Children, USAID, JSI/L10K, Ethiopian Pediatrics Society, the Federal Ministry of Health, Regional Health Bureaus, etc. The WHO PSBI management guideline and the evidence that contributed to its development [12–16] were presented and discussed. Additional policy dialogue sessions were conducted at regional, zonal and district levels, mainly by the TSU, to ensure understanding of evidence and implications for implementation. Following this activity, the Oromia RHB together with the Jimma site study team identified potential sites/districts for the study.

**2. Agreement on identification and management of PSBI cases.** During the policy dialogue workshops, an agreement was reached on the management of sick young infants with signs of PSBI at health posts and health centres. The HEWs would identify sick young infants in the community or at the health post, assess and classify for PSBI, refer them to health centres when required, and treat and follow up those who do not require a referral or whose caregivers refuse referral. Those infants whose care givers accept referral by the HEW would go to health centres to be reassessed and referred to a hospital if needed. At both levels, when a referral is refused, treatment would be provided according to the agreed-upon standards as shown in Box 2.

## 3. Preparatory phase

*i. Establishment of a Technical Support Unit (TSU).* A TSU composed of three paediatricians, one reproductive health expert, one microbiologist and one sociologist (all from Jimma University) and one newborn health programme manager (from Oromia RHB) was established. To coordinate field activities, one full-time coordinator was based in each district. The coordinators provided technical support and mentored the HEWs, validated a selection of enrolled cases, assessed the outcome of treated sick young infants, collected quantitative data from the health posts and health centres and coordinated the overall study-related activities.

The roles and responsibilities of the TSU were to: a) develop an implementation plan and data collection instruments; b) prepare the study sites; c) arrange and participate in stakeholders' meetings before and during the research; d) train health care providers at health posts and health centres as well as managers; e) conduct monthly supportive supervision at health posts and health centres through performance assessment and feedback; f) assess the performance of HEWs and health centre staff and provide feedback; g) identify implementation challenges and develop interventions in collaboration with stakeholders; h) compile health post and health centre data, and i) supportcommunity sensitization.

The TSU established effective communication between the study team and other stakeholders, such as JSI/L10K, zonal and district health offices, and health care providers at health posts and health centres. The district MCH coordinators oversaw the districts' MCH activities. They conducted monthly supportive supervision of the health centres and selected health posts, compiled data from the health facilities and submitted it to the TSU.

*ii. Roles and responsibilities of the nongovernmental organization.* JSI/L10K facilitated the initial training of the HEWs, provided the initial start-up commodities and supplies necessary to assess, classify and treat sick young infants, replenished these commodities and supplies for some of the health posts, carried out supportive supervision to the health posts and conducted quarterly review meetings with the HEWs, the TSU and the district health office.

*iii. Roles and responsibilities of the government health department.* The RHB led implementation in coordination with the district officers and took part in the review meetings as well as the supportive supervision, where progress and lessons learned were discussed.

*iv. Building health system capacity.* Training: The TSU and JSI/L10K trained all the HEWs and health centre staff in the study districts in June 2016. Subsequently, refresher training was

given with a focus on gaps identified during regular TSU and JSI/L10K supportive supervision.

Tools, job aids and equipment: Tools and job aids, developed by the Federal Ministry of Health (FMOH) (CBNC chart booklet, CBNC register and family health booklet), were provided to the HEWs to support and facilitate their work. They were equipped with thermometers, respiratory rate counters and weighing scales.

*v. Community mobilizations.* To create awareness and mobilize the community, we used the existing local government structures focusing mainly on the HEWs and the HDA. The major platforms used to mobilize the communities were the health extension workers (HEW) and health development army (HDA) linkages so that most deliveries and sick young infants are identified. Various community gatherings/meetings to deliver the necessary key messages about identification and treatment of sick young infants and the campaigns organized by the PHCU including the community health insurance, the tuberculosis screening, trachoma control, onchocerciasis control etc. These campaigns were used to disseminate the necessary information to the community when they gathered so that they could contribute to the implementation research. We have also included this under the methods section.

*vi. Ethical clearance.* Ethical clearance for this implementation research was obtained from Jimma University Institutional Review Board and the WHO Research Ethics Review Committee. Additionally, letters of support were obtained from the respective national, regional, zonal and district government offices. Individual participant's informed consent was waived since the implementation research was done in the routine government health system.

## 4. Implementation phase

*i. Logistics and commodities.* Logistics management for the health posts was handled by JSI/L10K, which provided 50 vials of gentamicin and 150 strips of oral amoxicillin dispersible tablets (10 tablets per strip) for each health post at the beginning of the study. Disposable syringes (3ccs) were also provided. The district health offices provided subsequent supplies through the routine delivery system. Although medicines at health posts were available free of charge for PSBI cases, at the health centres caregivers were expected to pay for them.

The TSU was not involved in the procurement and distribution of supplies but shared observations about the availability and utilization of commodities with the district health offices and JSI/L10K. The estimation of needs and distribution was based on the annual number of live births in the catchment area of a health post using an estimated birth rate and the number of women of reproductive age, and assuming 10% of live births would develop PSBI in the first two months of life. Logistics management at the district level was carried out by the MCH coordinator, and by pharmacy personnel and HEWs at the health centre and health post level, respectively.

*ii. Supervision.* Supportive supervision was conducted at two levels. First, health professionals from the health centres conducted supportive supervision every two weeks to the health posts in their catchment area. During supervision, HEWs and supervisors discussed issues that were challenging for the HEWs about identification, classification and treatment of sick young infants. On-the-job training, case discussion and assessment of HEWs' knowledge, skills and practices were part of the supervision. Second, team members from the TSU conducted monthly supportive supervision for the study districts, health centres and health posts. During district-level supervision, MCH coordinators and supervisors from selected health facilities also joined the team.

*iii. Quality control.* The performance of individual health centre staff and HEWs were reviewed during monthly supportive supervision and quarterly review meetings, and necessary

feedback was provided. The parameters used during this assessment included the number of sick young infants identified, assessed and treated; the quality of assessment and treatment; the quality of record-keeping; the number of pregnancies and births identified; and the number of postnatal care visits. Additionally, the performance was compared among health facilities and feedback given. Solutions to identified implementation barriers were developed collaboratively. Better performers were encouraged to sustain and enhance their performance and share their experiences with other facilities during review meetings.

## Results

Between September 2016 and August 2017, 6185 live births were recorded and visited by HEWs in the study area. A total of 601 young infants 0–59 days of age were identified with signs of PSBI either at health posts (n = 426, 71.0%) and health centres (n = 175, 29.0%). Assuming that 25% of births were missed (expected births = 7731) and around 10% of births were expected to have an episode of PSBI in the first two months of life [15, 20] (expected number of PSBI cases = 773), the coverage of appropriate treatment for PSBI was 77.7% (601/773).

Of the 601 infants with PSBI, 432 (71.9%) infants 7–59 days old had fast breathing only pneumonia, whereas 155 (25.8%) and 14 (2.3%) infants had CSI or critical illness, respectively. Of the 432 infants, 7–59 days of age with fast breathing only, 321 (74.0%) presented at health posts and received treatment; the remaining infants were seen and treated at the health centres (Table 1). At both levels, the treatment was with twice-daily oral amoxicillin for seven days. Of these 432 infants, 412 (95.0%) completed all 14 doses of oral amoxicillin. There were no deaths among infants presenting with fast breathing pneumonia.

Among 169 infants who either had CSI (n = 155) or critical illness (n = 14), only about two thirds (n = 110, 65.0%) were referred; 64 (38.0%) from health posts to health centres, and 46 (27.0%) from health centres to hospitals. The remaining infants (n = 59, 35.0%) were treated as outpatients without being offered referral. The referral acceptance rate was 9.4% (6/64) from health posts to health centres and 52.2% (24/46) from health centres to hospitals. All the infants whose caregivers refused referral accepted treatment at either the health posts or health centres (Fig 1).

Of all sick young infants treated at the health posts and health centres, 129/132 (98.0%) of those with CSI and all of those with critical illness showed improvement after completing their

**Table 1. Infants 7–59 days old with fast breathing onlypneumonia treated on an outpatient basis without a referral (N = 432).**

| Parameter | Heath post n (%) n = 321 | Health centre n (%) n = 111 | Total n (%) N = 432 |
|---|---|---|---|
| **Compliance with treatment** | | | |
| Received all 14 doses of amoxicillin | 313 (97.5) | 99 (89.2) | 412 (95.4) |
| Missing data | 8 (2.5) | 12 (10.8) | 20 (4.6) |
| **Follow-up of infants** | | | |
| Completed day 14 follow-up visit | 313 (97.5) | 99 (89.2) | 412 (95.4) |
| Partial follow-up (did not complete day 14 follow-up visit) | 8 (2.5) | 12 (10.8) | 20 (4.6) |
| **Treatment outcomes** | | | |
| Clinical treatment success | 318 (99.0) | 101 (91.0) | 419 (97.0) |
| Lost to follow-up | 3 (1.0) | 10 (9.0) | 13 (3.0) |
| Deaths | 0 | 0 | 0 |

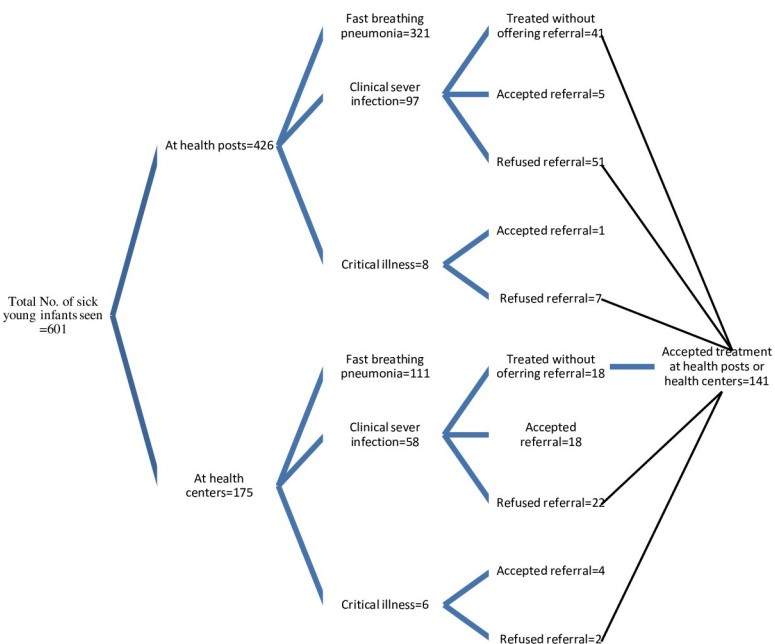

**Fig 1. Number of infants with any sign of PSBI assessed, classified and managed at different levels of health facility.**

treatment (Table 2). All nine infants with critical illness treated on an outpatient basis completed the prescribed doses of injectable gentamicin and oral amoxicillin, because injectable ampicillin was not available at the health posts and those at health centres didn't follow the recommended guideline. One infant with signs of CSI who was treated at a health centre for four days developed convulsions (an indicator of treatment failure) and was referred to hospital where he was cured after 14 days of inpatient treatment. Four sick young infants, two with signs of critical illness and two with CSI, who were referred died on the way before reaching a hospital, whereas two infants with critical illness referred from the health centre to a hospital completed their inpatient treatment and improved.

In Tiro Afata District where the two-day gentamicin plus seven-day oral amoxicillin regimen was followed, 74 CSI cases were treated, only one case of treatment failure was reported. In Gera District, where the seven-day gentamicin plus seven-day oral amoxicillin regimen was followed, 58 infants with CSI were treated and no case of treatment failure was seen. The compliance to treatment and treatment outcomes in both districts were similar (Table 3).

## Coordination and collaboration between the TSU and stakeholders

The commitment of the FMOH, RHB, the zonal and district health offices and their close collaboration with the implementing partner and the TSU was noteworthy. From the beginning to completion of the study, the TSU worked closely with these stakeholders regarding the various aspects of implementation and shared progress, challenges and suggestions with them. TSU, the district health offices and JSI/L10K jointly conducted review meetings, training and supportive supervision, in addition to the supportive supervision and meetings the district health offices held with HEWs and health centre staff. These joint learning platforms facilitated the integration of PSBI implementation into routine district health office activities. Additionally, at the national level, apart from providing preliminary findings quarterly to the FMOH, the TSU presented the findings in several workshops organized by the Ministry of Health and

**Table 2. Identification and outcomes of infants 0–59 days old with signs of clinical severe infection presenting at health centres and health posts.**

| Parameter | Level of health facility | | Total n (%) |
|---|---|---|---|
| | Health post n (%) n = 97 | Health centre n (%) n = 58 | N = 155 |
| Treated without being offered a referral to next-level health facility* | 41 (42.3) | 18 (31.0) | 59 (38.0) |
| Referred to next-level health facility | 56 (57.7) | 40 (68.9) | 96 (62.0) |
| Refused to accept referral advice to the next-level health facility | 51 (52.6) | 22 (37.0) | 73 (47.0) |
| Accepted referral advice to the next-level health facility | 5 (5.2) | 18 (31.0) | 23 (14.8%) |
| Accepted and received outpatient treatment at the initial health facility | 92 (94.8) | 40 (68.9) | 132 (85) |
| **Compliance with treatment on an outpatient basis** | | | |
| Completed treatment | 90 (92.8) | 40 (68.9) | 130 (83.9) |
| Completed all the respective 2 or 7 injections of gentamicin | 90 (92.8) | 40 (68.9) | 130 (83.9) |
| Received all 14 doses of oral amoxicillin | 90 (92.8) | 38 (65.6) | 128 (82.6) |
| Missing data | 2 (2.1) | 2 (3.4) | 4 (2.6) |
| **Follow-up of infants treated on an outpatient basis** | | | |
| Completed day 14 follow-up visit | 90 (92.8) | 38 (65.6) | 128 (82.6) |
| Partially followed-up (did not complete day 14 follow-up visit) | 1 (1.0) | 1 (1.7) | 2 (1.3) |
| Lost to follow-up (outcome unknown) | 1 (1.0) | 1 (1.7) | 2 (1.3) |
| **Treatment outcomes** | | | |
| Declared as treatment success | 91 (93.8) | 38 (65.6) | 129 (83) |
| Declared as 'Clinical treatment failure'† | 0 | 1 (1.7) | 1 (1.3) |
| **The outcome of the illness** | | | |
| 'Better' (including those treated at hospital) ‡ | 91 (93.8) | 38 (65.6) | 129 (97.7) |
| 'Still sick' | 0 | 1 (1.7) | 1 (0.65) |
| Outcome unknown | 0 | 1 (1.7) | 1 (0.65) |
| Died¶ | 0 | 2 (3.6) | 2 (1.3) |

*Did not follow the guideline recommendation.

†Developed convulsions on the 4th day of treatment for CSI and referred to a hospital; improved and discharged after inpatient treatment.

‡Additionally, 22 infants, 21 from among those who had accepted referral to a hospital after initial identification and one who was referred after failing simplified treatment at a health centre improved after completing their inpatient treatment.

¶Both deaths occurred in infants who were referred from a health centre to the referral hospitalbut died before reaching there.

its partners during and after the completion of the implementation research, to disseminate the results. Moreover, the TSU presented the findings to MCH Directorate of FMOH and the Research Advisory Council established under this directorate so that the findings could be translated into policy and guidelines.

## Implementation challenges and solutions

Throughout project implementation, the TSU worked hand-in-hand with the district and regional programme implementers to provide technical back-up, identify barriers to

**Table 3. Infants 0–59 days old with clinical severe infection treated at a health post or health centre with two-day or seven-day injectable gentamicin regimen.**

| Parameter | Gentamicin regimen | | Total n (%) N = 132 |
|---|---|---|---|
| | 2 days n (%) n = 74 | 7 days n (%) n = 58 | |
| **Accepted treatment at the primary health facility with a simplified antibioticregimen** | | | |
| Completed all injections | 74 (100.0) | 56 (96.6) | 130 (98.5) |
| Completed all 14 doses of amoxicillin* | 72 (97.3) | 56 (96.6) | 128 (97.0) |
| **Treatment outcomes** | | | |
| Declared as treatment success | 73 (98.6) | 56 (96.6) | 129 (96.7) |
| Declared as 'Clinical treatment failure'† | 1 (1.4) | 0 | 1 (0.8) |
| Refused treatment at the first-level health facility | 0 | 0 | 0 |

*One infant discontinued treatment after receiving two injections, and the other after four injections.

†Developed convulsions on 4th day of treatment for CSI and referred to hospital, improved and discharged after inpatient treatment.

implementation and potential solutions. We encountered several challenges during the implementation research and identified solutions in collaboration with stakeholders at all levels and at different times (Table 4). Although we managed to overcome most of the challenges, a few remain that will require the attention of policy-makers at all levels. The challenges observed were similar across both districts.

## Discussion

Our findings demonstrate that it is possible to implement the WHO guideline on PSBI management when the referral is not feasible [12] within existing programme settings in Ethiopia. Trained and supervised HEWs and nurses in PHCUs were able to identify and treat 7–59 days old infants with only fast breathing pneumonia without referral with oral amoxicillin, and sick young infants with signs of clinical severe infection and critical illness on an outpatient basis with simplified antibiotic regimen when the referral was not feasible. This finding is in line with reports from other PSBI implementation research studies in Mekelle, Ethiopia (Leul A, Mekelle University, personal communication); Kushtia, Bangladesh [21]; MaMoni project Bangladesh [22]; Lucknow, India [23]; Malawi [24] and Zaria, Nigeria [25].

Our treatment coverage of 78% was higher than that reported from other PSBI research sites from Kushtia, Bangladesh (31%) [21]; Lucknow, India (53%) [23]; and Malawi (64%) [24], but lower than Zaria, Nigeria, 96% [25]. Most families refused referral advice (76%) and accepted outpatient treatment with oral amoxicillin and gentamicin, a finding consistent with other studies from Africa and Asia; Zaria, Nigeria (97%) [25]; Ntcheu, Malawi (93%) [24]; Sylhet and Lakshmipur, Bangladesh (83%) [26]; and Lucknow, India (81%) [23]. We found high treatment completion rates, which we believe are mainly due to the counselling skills of health workers and acceptance of simplified treatment by families with additional support from the TSU to the health workers. Similar findings have been reported by MaMoni project, Bangladesh (80%) [27]; Malawi (95%) [24]; and Zaria, Nigeria (94.1%) [25] in infants with clinical severe infection who received outpatient treatment returned to the facility for the second antibiotic injection.

Our data showed higher treatment completion, low treatment failure and low mortality in infants treated at PHCUs after the refusal of referral, which has also been reported in other studies from Africa and Asia [21, 24, 25]. Given these positive results, we believe our intervention contributed to saving many newborn lives. A community-based PSBI management intervention trial conducted in rural Ethiopia also showed a 17% reduction in post-day 1 neonatal mortality [28].

**Table 4. Challenges faced during implementation of PSBI intervention and solutions.**

| Challenges | Solutions |
|---|---|
| Community's lack of awareness about service availability at the health post level and community perception that HEWs were not able to take care of sick young infants. | Use of radio services (in local language) to inform about service availability. HEWs used different community fora/gatherings to disseminate information about the availability of services for sick young infants at health posts. This led to an improvement in the utilization of services by the community. |
| Weak community networks–HDA networks were supposed to play key roles in mobilizing the community and assisting HEWs in the identification of pregnancies, births and sick young infants, but did not do so. In most cases, HDAs were unaware that HEWs were able to manage sick young infants. However, they did relatively better in the identification of pregnancies and births than the identification of sick young infants. | Regular meetings and discussions with the district health offices were used to stress the importance of strengthening and monitoring of community networks. This remains a challenge which needs further work from the community and the local government to improve and sustain the coverage of CBNC. |
| Unavailability of HEWs and closure of health posts-during random visits, we found that some health posts were closed. Some of the reasons included: a single HEW assigned to the health post was on anassignment elsewhere; even when there was more than one HEW, at times all of them went out at the same time for other activities (campaigns, meetings, trainings); HEWs lived far from the health post; and HEWs were away for social or personal reasons, including maternity leave. | Discussions were held with local health authorities, health centre staff and HEWs to consider task shifting and also to increase the number of HEWs. Although there were some improvements, this challenge remained throughout the study period, requiring serious consideration and commitment from all stakeholders. Local, regional and national political leaders, as well as the multiple nongovernmental and governmental organizations working on the health extension programme, need to be actively involved in addressing this issue to ensure the uninterrupted availability of services at health posts. |
| Inappropriate management of sick young infants—during the supportive supervision visits, especially during the first few months of implementation, both HEWs and health centre staff were observed to have deficiencies in managing sick young infants. This included treating at the health post even when a referral was indicated. | TSU and the implementing partner provided close supportive supervision to the HEWs and health centre staff during which cases were reviewed and feedback was given. Review meetings and additional training were also provided. Through this continuous support and feedback, the situation improved substantially, but continuous mentoring and support will be required to have a sustainable effect. |
| Stock-out of medicines for the management of PSBI in some health facilities for a few months was observed. | When informed, JSI/L10K re-filled the medications for the management of PSBI until the district health offices took over. |
| Poor documentation, including incomplete or missing data, on clinical assessment, treatment provided and follow-up appointments were noted. | The TSU provided additional skills-building and refresher training to HEWs and health centre staff. Jointly with the district health office, we provided continuous supportive supervision and conducted review meetings during which we reviewed the documentation of each health facility and provided necessary corrective feedback. This led to an improvement in the documentation at the health facilities. |
| Less attention given to health centres and poor linkage between health posts and health centres–staff not trained in WHO PSBI management guideline, different treatment guideline used at the health centre creating confusion and disagreements when health centre staff were supervising HEWs, and during the referral of sick young infants from health posts to health centres. | In collaboration with the JSI/L10K and the district health offices, the TSU facilitated the harmonization of the management protocol for PSBI used at the two levels and provided the necessary training and supportive supervision to health centre staff. These interventions improved the alignment between the two levels of care. |

We reported a much higher rate of fast breathing pneumonia in 7–59 day-old infants (72%) than that reported in some other studies (30% in MaMoni project, Bangladesh, 28% in Malawi, 22% in Zaria, Nigeria and 13.3% in Lucknow, India) [22–25], but similar to that documented in Tigray Region, Ethiopia (Leul A, Mekelle University, personal communication), but lower than that reported from Kushtia, Bangladesh (87%) [21]. The potential reasons for the high

proportion of fast breathing pneumonia in Ethiopia compared to other countries are that counting respiratory rate in children under 5 years of age is a routine practice that has been introduced as part of iCCM/CNBC and Integrated Management of Newborn and Child Illness in Ethiopia for the last several years. Thus, health care providers are well trained and equipped with respiratory rate timers to count respiratory rate as a routine practice for all children presenting to a health post or health centre. These infants were successfully treated without a referral, as reported by other studies [21–25], making this an important strategy for increasing access to care, thereby further contributing to the reduction of young infant deaths in low- and middle-income countries.

Although the implementation of the simplified treatment of PSBI when a referral is not feasible has created an opportunity for these sick young infants who otherwise might have not been treated, the high rate of referral refusal and the associated reasons are issues that need urgent attention. Applegate and colleagues [27] identified economic and household factors, as well as previous experiences with poor quality of care at the sub-district hospital, as major barriers to referral acceptance in rural Bangladesh, whereas Guenther and colleagues identified a poor referral system, specifically lack of transport, in Malawi for lower acceptance of referral [24].

Over one-third of the infants with CSI and critical illness were assessed at health centres, which serve as the referral site for health posts. They often have more staff with higher qualifications, and yet in our study, the same treatment protocol was followed at both the health centre and health post levels. If health centres are the referral centres for health posts, then they should be strengthened to provide additional care, particularly for infants presenting with a more serious illness. Health centres do have a few beds (for mothers in labour/delivery rooms, not for newborns), oxygen concentrators and neonatal warmers (although electricity interruptions may occur) and health care staff round the clock.

In our study, the families of nine out of the 14 infants with critical illness refused referral and two infants with critical illness died on the way to the hospital. We could not validate the diagnosis of these young infants with a critical illness, who ideally should have been treated at a hospital. In the absence of hospitalization at least they should have been treated with injectable gentamicin plus injectable ampicillin for up to 7 days (Box 2) [12], but they were all treated with injectable gentamicin and oral amoxicillin, which was non-adherence of the guideline. It was understandable at the health post because injectable ampicillin was not available there, but we are unsure why health centres treated such patients with oral amoxicillin when families refused referral advice.

We observed that a substantial number of infants with signs of CSI who should have been advised for a referral to a higher facility received outpatient treatment without referral advice. Non-adherence to the recommended guideline by health care providers was also documented in the two PSBI implementation research studies from Bangladesh [21, 22]. We discussed this issue with health workers during the supervisory visits and tried to find the reasons behind this non-adherence, which included i) a perception of health workers/health care providers that they can treat all the infants without a referral, and ii) their assumption that some families would not accept referral even when offered. The importance of referral when indicated and implementation of the standard PSBI protocol were reinforced by the TSU and programme implementers during training and supervisory visits. This situation improved with time, as was also reported by Applegate et al from Bangladesh [22].

Concerning the two-day and seven-day gentamicin injection regimen, we did not find any difference in treatment adherence, completion rate or success rate, but the sample size was too small to draw any firm conclusions. From the practical health care system point of view, the two-day injectable therapy regimen appears to bemore attractive.

Many of the implementation challenges encountered, including the unpredictable availability of HEWs at health posts and the interrupted supply of commodities, have also been documented in previous studies that looked into the effectiveness of the health extension programme and other community-based services in Ethiopia [17, 29]. These systemic issues need urgent action by policy-makers.

A significant policy change has occurred as a result of this implementation research at our site and in the Tigray region, Ethiopia. In 2018, the Government of Ethiopia revised the chart booklet for integrated community management of newborn and childhood illness used for iCCM and CNBC [30], which is consistent with the WHO PSBI guideline [12]. Young infants 7–59 day-old with only fast breathing are treated with oral amoxicillin for seven days without referral and those with signs of clinical severe infection are treated with injectable gentamicin for two days plus oral amoxicillin regimen for seven days at primary health care centres when referral is not feasible.

The strengths of our implementation research were the multidisciplinary approach used, close supportive supervision of health facilities, close collaboration between the TSU and the local, regional and national level authorities and the fact that it was done within the health system using existing infrastructure and personnel.

Our study had some limitations. First, as the referral hospitals were outside the study district, due to logistical reasons we were unable to capture information on sick young infants who were either taken directly to hospitals or were born in the hospital and admitted with signs of PSBI. In a parallel study in Tigray Region, out of 850 infants with PSBI, about 330 infants with PSBI were directly taken to hospitals (Leul A, Mekelle University, personal communication). Second, we missed recording around one-quarter of livebirths. Both the above-mentioned limitations resulted in a slightly lower coverage of PSBI management than anticipated. Third, it was not possible to do on-the-spot verification or validation of the performance of the HEWs and health centre staff about management of each sick infant, as it was not easy to reach every health facility each time a sick young infant was seen. Hence, there may be some misclassification and inappropriate treatment of the cases, which was also reported from Bangladesh [27]. Finally, a potential limitation could be that our good treatment outcomes were achieved through the strong and close support provided by the TSU as reported from Zaria, Nigeria and Lucknow India [23, 25], and might not reflect the actual situation in the absence of such support under the routine health care system.

In conclusion, the management of sick young infants with PSBI at the primary health care level when the referral is not feasible is acceptable and feasible in this setting in Ethiopia, contributing to high treatment coverage and thereby saving many newborn lives. Although we addressed different challenges encountered in collaboration with the stakeholders, some such as weak community networks, inadequate attention given to health centres, and HEWs not always in their assigned posts remained at the time of completion of the research. Other challenges could re-appear if continuous mentoring and supportive supervision are not sustained. Hence, all the responsible stakeholders, including the community, district and zonal health offices, RHB, FMOH, regional and national political leaders as well as the multiple nongovernmental organizations working with the PHCUs should make substantial efforts to ensure the sustainability of community-based interventions and the coordination of various activities to optimize care of young infants in Ethiopia.

## Acknowledgments

We thank the FMOH of Ethiopia, Oromia RHB, JSI/L10K as well as Jimma Zone and Tiro Afata and Gera District Health Offices for all their support to this implementation research.

We also thank all the HEWs, the health centre staffs, community members and all who assisted the implementation research in various ways.

## Author Contributions

**Conceptualization:** Melkamu Berhane, Tsinuel Girma, Workneh Tesfaye, Nega Jibat, Mulumebet Abera, Samira Aboubaker, Yasir Bin Nisar, Shamim Ahmad Qazi, Rajiv Bahl, Alemseged Abdissa.

**Data curation:** Melkamu Berhane, Tsinuel Girma, Workneh Tesfaye, Nega Jibat, Mulumebet Abera, Sufian Abrahim, Alemseged Abdissa.

**Formal analysis:** Melkamu Berhane, Alemseged Abdissa.

**Funding acquisition:** Samira Aboubaker, Yasir Bin Nisar, Shamim Ahmad Qazi, Rajiv Bahl.

**Investigation:** Melkamu Berhane, Tsinuel Girma, Workneh Tesfaye, Nega Jibat, Mulumebet Abera, Sufian Abrahim, Samira Aboubaker, Yasir Bin Nisar, Shamim Ahmad Qazi, Rajiv Bahl, Alemseged Abdissa.

**Methodology:** Samira Aboubaker, Yasir Bin Nisar, Shamim Ahmad Qazi, Rajiv Bahl, Alemseged Abdissa.

**Project administration:** Melkamu Berhane, Alemseged Abdissa.

**Supervision:** Melkamu Berhane, Tsinuel Girma, Nega Jibat, Mulumebet Abera, Sufian Abrahim, Samira Aboubaker, Yasir Bin Nisar, Shamim Ahmad Qazi, Alemseged Abdissa.

**Writing – original draft:** Melkamu Berhane.

**Writing – review & editing:** Melkamu Berhane, Tsinuel Girma, Workneh Tesfaye, Nega Jibat, Mulumebet Abera, Sufian Abrahim, Samira Aboubaker, Yasir Bin Nisar, Shamim Ahmad Qazi, Rajiv Bahl, Alemseged Abdissa.

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
