## [Decision Letter · Decision Letter 0]

26 Oct 2020

PONE-D-20-25280

Implementation research on management of sick young infants with possible serious bacterial infection when referral is not possible in Jimma Zone, Ethiopia: Challenges and solutions

PLOS ONE

Dear Dr. Berhane,

Thank you for submitting your manuscript to PLOS ONE. After careful consideration, we feel that it has merit but does not fully meet PLOS ONE’s publication criteria as it currently stands. Therefore, we invite you to submit a revised version of the manuscript that addresses the points raised during the review process.

We look forward to receiving your revised manuscript.

Kind regards,

Elizeus Rutebemberwa

Academic Editor

PLOS ONE

Journal Requirements:

2.Please provide additional details regarding participant consent. In the ethics statement in the Methods and online submission information, please ensure that you have specified (1) whether consent was informed and (2) what type you obtained (for instance, written or verbal, and if verbal, how it was documented and witnessed). If your study included minors, state whether you obtained consent from parents or guardians. If the need for consent was waived by the ethics committee, please include this information.

3.Thank you for stating the following in the Competing Interests section:

[The authors have declared that no competing interest exists. Rajiv Bahl and Yasir Bin Nisar are staff members of the World Health Organization. The expressed views and opinions do not necessarily express the policies of the World Health Organization.].

Reviewers' comments:

Reviewer's Responses to Questions

**Comments to the Author**

1. Is the manuscript technically sound, and do the data support the conclusions?

Reviewer #1: Yes

Reviewer #2: Yes

2. Has the statistical analysis been performed appropriately and rigorously? 

Reviewer #1: Yes

Reviewer #2: Yes

3. Have the authors made all data underlying the findings in their manuscript fully available?

Reviewer #1: No

Reviewer #2: Yes

4. Is the manuscript presented in an intelligible fashion and written in standard English?

Reviewer #1: No

Reviewer #2: Yes

5. Review Comments to the Author

Reviewer #1: General comment: The paper needs language polishing because it is hard to read in its current form.

For instance: The implementation research was carried out in phases.

Statistical analysis:

The used descriptive statistics seem appropriate.

Reviewer #2: This paper is highly commendable, not only because it addresses an important challenge in public health in resource poor settings, but also because it gives an insight in essential aspects of implementation research! My comments are thus mainly to the implementation research methodology and less to details in the specific findings.

A fundamental issue is the participatory approach. How was stakeholders defined? The hierarchy of the health architecture is mobilized as well as various cadres of health workers. But it is not so clear whether and how civil society organizations and the population was mobilized as co-workers. HEWs is a special invention in Ethiopia, and it would have been nice to see a broader description of their role and function, not least to explain some of the problems encountered (health posts closed, stock-outs etc), but also the success. The poor function of the HDAs is another finding that is difficult to understand without some more explanation of their recruitment and support from the authorities.

Sustainability of the program seems to have been high on the agenda in planning this project. More details about how much extraordinary resources were mobilized from sponsors (BMGF and WHO) and how the future gap is meant to be filled.

The TSUs have given ample emphasis to quality control during their supportive supervisions, but the paper gives little details about the findings. How correctly were PSBI / CSI / critical cases diagnosed? What indicators for job satisfaction were collected?

One specific finding that seems to be almost obligatory in implementation research is that health workers find recording and reporting burdensome and often fail to comply with the rules. This is such a fundamental threat to the quality of the research that I would have liked to have the authors’ ideas about how this challenge might be handled. Are we too slow to utilize new e-health tools?

Another finding that is very common in implementation research is problems about refusals and peoples’ lack of compliance with the organization of the health system. A number of clients bypass the level of health service they were meant to consult. Or, as in this study, they refuse referral to the next level. In the discussion the authors cite the findings in other studies, but I am not able to see what they found in this study. It is rather essential that the authorities understand why people make choices deviating from the plan.

My last comment to the implementation research method applied in this paper is that a plan for dissemination is lacking. The general structure of a scientific paper does not directly invite for this. But the whole purpose of implementation research is to find out whether the findings give a basis for adjusting the program under study. While operations research (or quality assurance) is asking: Are we doing things right (according to the guidelines), implementation research is asking: Are we doing the right things (do we need to change the guidelines)? This plan for feed-back to the program itself is to me the very essence of implementation research. I would be very happy to see some reflections on this in the revised manuscript.

6. PLOS authors have the option to publish the peer review history of their article (what does this mean?). If published, this will include your full peer review and any attached files.

Reviewer #1: No

Reviewer #2: **Yes: **Gunnar Bjune

---

## [Author Response · Author response to Decision Letter 0]

21 Nov 2020

Pont by point responses to Editor’s and reviewers’ comments and suggestions

Thanking the editor and reviewers for their valuable comments and suggestions which are helpful to improve the quality of our manuscript, we hereby provide point by point responses to the comments and suggestions.

Editor’s comments and suggestions:

• Comment/suggestion: Please ensure that your manuscript meets PLOS ONE's style requirements, including those for file naming.

Our response: We have made the necessary corrections accordingly.

• Comment and suggestion: Please provide additional details regarding participant consent. In the ethics statement in the Methods and online submission information, please ensure that you have specified (1) whether consent was informed and (2) what type you obtained (for instance, written or verbal, and if verbal, how it was documented and witnessed). If your study included minors, state whether you obtained consent from parents or guardians. If the need for consent was waived by the ethics committee, please include this information.

Our response: We have provided accordingly.

• Comment/suggestion: Thank you for stating the following in the Competing Interests section:

[The authors have declared that no competing interest exists. Rajiv Bahl and Yasir Bin Nisar are staff members of the World Health Organization. The expressed views and opinions do not necessarily express the policies of the World Health Organization.]. 

Please confirm that this does not alter your adherence to all PLOS ONE policies on sharing data and materials, by including the following statement: "This does not alter our adherence to PLOS ONE policies on sharing data and materials.” (as detailed online in our guide for authors http://journals.plos.org/plosone/s/competing-interests ). If there are restrictions on sharing of data and/or materials, please state these. Please note that we cannot proceed with consideration of your article until this information has been declared.

Our response: We have included it accordingly.

Reviewer #1: 

• General comment: The paper needs language polishing because it is hard to read in its current form. For instance: The implementation research was carried out in phases.

Our response: We have revised the language.

Reviewer #2: 

• Comment: A fundamental issue is the participatory approach. How was stakeholders defined? The hierarchy of the health architecture is mobilized as well as various cadres of health workers. But it is not so clear whether and how civil society organizations and the population was mobilized as co-workers.

Our response: We have elaborated these in the methods section. 

i) Stakeholders were defined as individuals, group of individuals, professional societies, governmental and non-governmental organizations working on newborn and child health at the national and/or sub-national levels. As we explained in the methods part, these stakeholders’ participation was ensured at different levels (national and sub-national) and different phases of the implementation research and its outcomes such as the orientation and policy dialogue, workshops at the launch, and at mid-term and end, the dissemination workshops, and translation of findings into policy and guidelines meetings. Similarly, the civic societies like the Ethiopian Pediatrics Society were involved in different phases of the implementation (line 190 to 198). 

ii) To mobilize the community, we used the existing local government structures. The major platforms used to mobilize the communities were the health extension workers (HEW) and health development army (HDA) linkages so that most deliveries and sick young infants are identified. Community gatherings/meetings including campaigns organized by the primary health care unit (PHCU) like the community health insurance, the tuberculosis screening, trachoma control, onchocerciasis control were used to deliver the necessary key messages about identification and treatment of sick young infants and support the implementation research. We have also included this under the methods section (line 248 to 258).

• Comment: HEWs is a special invention in Ethiopia, and it would have been nice to see a broader description of their role and function, not least to explain some of the problems encountered (health posts closed, stock-outs etc), but also the success.

Our response: We have presented this in the methods section. We have also added additional information as per the comment and suggestion (Panel 1 page 7 to 9). 

• Comment: The poor function of the HDAs is another finding that is difficult to understand without some more explanation of their recruitment and support from the authorities.

Our response: We have added this in the methods section (Panel 1 page 9 to 10). 

• Comment: Sustainability of the program seems to have been high on the agenda in planning this project. More details about how much extraordinary resources were mobilized from sponsors (BMGF and WHO) and how the future gap is meant to be filled.

Our responses: 

i) Yes, it is true that the sustainability of the implementation is a high priority issue. Especial emphasis needs to be given to the mentoring and supportive supervision of the PHCUs so that identification, accurate assessment, classification and treatment of the sick young infants can sustainably be carried out. Given that the implementation research was done in the existing government health system, the program can sustainably be carried out provided that the practices are supported by national policy and guideline, the necessary commodities are available, appropriate training is given to the health workers and the necessary mentoring and supportive supervision are provided to the PHCUs. However, since we are reporting what happened during the implementation research and we didn’t do any formal assessment then after, we can’t say anything whether this has been continued or no.

ii) The resource mobilized from the BMGF through the WHO was mainly used for the technical support unit (TSU), which was utilized for the supportive supervision, filling some training gaps, data collection, community mobilization/awareness creation activities and stakeholders’ meetings conducted locally. Otherwise, the initial training and the startup supplies/commodities were provided/supported by the government’s implementing partner whereas the subsequent supplies were provided by the local government system. Subsequently, as well, the local government is expected to take over all the logistics and other technical issues even if some non-governmental organizations might continue their support. We have indicated this in the discussion section (line 506 to 517).

• Comment: The TSUs have given ample emphasis on quality control during their supportive supervisions, but the paper gives little details about the findings. How correctly were PSBI / CSI / critical cases diagnosed? What indicators for job satisfaction were collected?

Our responses: Thank you for this valid comment which is an important aspect of such implementation research. We planned to carry out validation of sick young infants assessed, classified and managed at the PHCU at the beginning of the implementation research. However, as we have mentioned in the limitation of the study, we couldn’t do so due to difficulty in reaching every health facility every time an infant with PSBI/CSI was identified since the health facilities were very far apart. Though we were unable to do real-time verifications, during the supportive supervision visits we evaluated medical records for adherence to the guideline in terms of assessment, classification and treatment of cases. Additionally, we determined the adherence of the HEWs and health workers to the guideline/recommendation concerning the procedures to be followed, both of which were used as a proxy of correct assessment, classification and treatment of the sick young infants. With these, we have found that noteworthy proportions of sick young infants were found to be treated at the health centers and health posts without being offered referral, which we have indicated in the respective results and discussion parts (lines 321 to 324, 459 to 469, 498 to 502; Table 4, page 27). 

• Comment: One specific finding that seems to be almost obligatory in implementation research is that health workers find recording and reporting burdensome and often fail to comply with the rules. This is such a fundamental threat to the quality of the research that I would have liked to have the authors’ ideas about how this challenge might be handled. Are we too slow to utilize new e-health tools?

Our responses: Thank you again for this comment. As we have indicated on Table 4, we had this challenge especially in the first quarter of the implementation. We had a series of discussions with the health workers together with the district, zonal and regional health department personnel which led to substantial improvement. The TSU increased the frequency of its supportive supervision to the PHCU which has also contributed to the improvement. So, for similar implementation research to be carried out successfully, close collaboration with the local government authorities and other stakeholders to overcome such challenges is needed. We agree with the reviewer that e-health tools could potentially improve recording and reporting challenges.

• Comment: Another finding that is very common in implementation research is problems about refusals and peoples’ lack of compliance with the organization of the health system. A number of clients bypass the level of health service they were meant to consult. Or, as in this study, they refuse a referral to the next level. In the discussion, the authors cite the findings in other studies, but I am not able to see what they found in this study. The authorities must understand why people make choices deviating from the plan.

Our responses: Even if we found a high rate of refusal of referral, we didn’t try to assess the rate of bypassing the level of health facility they were meant to consult. Our target was to ensure that sick young infants are getting treatment at any level of the health facility. Therefore, we didn’t insist on the families to visit any specific level of health facility before the other, at the same time we did not encourage them to bypass one level for the other. And in fact, the community knew very well that they need to get a referral document from a lower-level health facility to get services from a higher-level facility and in general they abide by this principle. We also didn’t follow those patients who went directly to a hospital. We have mentioned the higher referral refusal rate in our results (line 324 to 327).

Comment: My last comment to the implementation research method applied in this paper is that a plan for dissemination is lacking. The general structure of a scientific paper does not directly invite for this. But the whole purpose of implementation research is to find out whether the findings give a basis for adjusting the program under study. While operations research (or quality assurance) is asking: Are we doing things right (according to the guidelines), implementation research is asking: Are we doing the right things (do we need to change the guidelines)? This plan for feed-back to the program itself is to me the very essence of implementation research. I would be very happy to see some reflections on this in the revised manuscript.

Our responses: We have presented the findings of this implementation study to the relevant stakeholders at different times given below (lines 374 to 380, 479 to 486).

i) Mid-term preliminary findings workshop was conducted after 6 months of the implementation in which various stakeholders participated (Federal Ministry of Health, Regional Health Bureau, UNICEF, WHO, Save the Children, Ethiopian Pediatrics Society, USAID, etc).

ii) At the end of implementation research, the findings were disseminated in a workshop in which several stakeholders took part.

iii) The investigators presented their findings to the Maternal, Newborn, Child and Adolescent Health and Nutrition (MNCAH-N) director and her team. 

iv) The results were also presented to the Research Advisory Council (RAC) of the Ministry of Health so that the findings of the study can be incorporated into the national policy and guidelines.

v) The research findings were presented at a scientific dialogue that was organized by Research Advisory Council (RAC) of MNCAH-N) Directorate that aimed to revise the national CBNC and iCCM guidelines. And indicated in the discussion of our manuscript, the findings together with findings from a sister study conducted in Mekelle in Tigray region, Ethiopia were used by the Ministry of Health to update the CBNC and iCCM guidelines in line with the current WHO recommendations/guidelines.

---

## [Decision Letter · Decision Letter 1]

1 Feb 2021

PONE-D-20-25280R1

Implementation research on management of sick young infants with possible serious bacterial infection when referral is not possible in Jimma Zone, Ethiopia: challenges and solutions

PLOS ONE

Dear Dr. Berhane,

Thank you for submitting your manuscript to PLOS ONE. After careful consideration, we feel that it has merit but does not fully meet PLOS ONE’s publication criteria as it currently stands. Therefore, we invite you to submit a revised version of the manuscript that addresses the points raised during the review process.

We look forward to receiving your revised manuscript.

Kind regards,

John S Lambert

Academic Editor

PLOS ONE

Reviewers' comments:

Reviewer's Responses to Questions

**Comments to the Author**

1. If the authors have adequately addressed your comments raised in a previous round of review and you feel that this manuscript is now acceptable for publication, you may indicate that here to bypass the “Comments to the Author” section, enter your conflict of interest statement in the “Confidential to Editor” section, and submit your "Accept" recommendation.

Reviewer #2: All comments have been addressed

2. Is the manuscript technically sound, and do the data support the conclusions?

Reviewer #2: Yes

3. Has the statistical analysis been performed appropriately and rigorously? 

Reviewer #2: Yes

4. Have the authors made all data underlying the findings in their manuscript fully available?

Reviewer #2: Yes

5. Is the manuscript presented in an intelligible fashion and written in standard English?

Reviewer #2: Yes

6. Review Comments to the Author

Reviewer #2: I appreciate the rather extensive description of the Ethiopian HEP and its components, but feel that this time it has become almost too extensive. Also the text contains a few wordings that indicate an evaluation of teh HEP without giving any reference to a scientfiic analysis of the program. Thus, I would like the authors to slim down the etxt describing HEP, but keep the essentials (that is a great help to understand the paper) and avoid value loaded sattements. otherwise I am fully satisfied with the revision.

7. PLOS authors have the option to publish the peer review history of their article (what does this mean?). If published, this will include your full peer review and any attached files.

Reviewer #2: **Yes: **Gunnar Bjune

---

## [Author Response · Author response to Decision Letter 1]

11 Feb 2021

We have included the respective references at the respective place. We have also modified the texts accordingly (panel 1).

---

## [Decision Letter · Decision Letter 2]

13 Jul 2021

Implementation research on management of sick young infants with possible serious bacterial infection when referral is not possible in Jimma Zone, Ethiopia: challenges and solutions

PONE-D-20-25280R2

Dear Dr. Berhane,

We’re pleased to inform you that your manuscript has been judged scientifically suitable for publication and will be formally accepted for publication once it meets all outstanding technical requirements.

Kind regards,

Judith Kose, M.D.

Academic Editor

PLOS ONE

Additional Editor Comments (optional):

Reviewers' comments:

Reviewer's Responses to Questions

**Comments to the Author**

1. If the authors have adequately addressed your comments raised in a previous round of review and you feel that this manuscript is now acceptable for publication, you may indicate that here to bypass the “Comments to the Author” section, enter your conflict of interest statement in the “Confidential to Editor” section, and submit your "Accept" recommendation.

Reviewer #2: All comments have been addressed

2. Is the manuscript technically sound, and do the data support the conclusions?

Reviewer #2: Yes

3. Has the statistical analysis been performed appropriately and rigorously? 

Reviewer #2: Yes

4. Have the authors made all data underlying the findings in their manuscript fully available?

Reviewer #2: Yes

5. Is the manuscript presented in an intelligible fashion and written in standard English?

Reviewer #2: Yes

6. Review Comments to the Author

Reviewer #2: I think this has now become a good paper that could serve as a model for many more to follow. In the future you might give more room for a chronology of events to show how the research followed and adjusted the implementation.

7. PLOS authors have the option to publish the peer review history of their article (what does this mean?). If published, this will include your full peer review and any attached files.

Reviewer #2: **Yes: **Gunnar Aksel Bjune

---

## [Editor Report · Acceptance letter]

19 Jul 2021

PONE-D-20-25280R2 

Implementation research on management of sick young infants with possible serious bacterial infection when referral is not possible in Jimma Zone, Ethiopia: challenges and solutions 

Dear Dr. Berhane:

I'm pleased to inform you that your manuscript has been deemed suitable for publication in PLOS ONE. Congratulations! Your manuscript is now with our production department. 

Kind regards, 

on behalf of

Dr. Judith Kose 

Academic Editor

PLOS ONE